# Perceived work-related stress and associated factors among the surgical workforce in a Nigerian tertiary health facility: A cross-sectional study

Jibril M. Bashar[1], Danjuma Aliyu[2], Emmanuel E. Anyebe[3], Israel Gabriel[4], Amanullahi Nasir[5], Abdulrrahaman S. Mangari[6], Faizah S. Abubakar[7], Yusuf H. Wada[5]*

1 Faculty of Clinical Sciences, Department of Community Medicine, Ahmadu Bello University, Zaria, Nigeria, 2 School of Post Basic Perioperative Nursing, Ahmadu Bello University Teaching Hospital, Tudun Wada Zaria, Kaduna State, Nigeria, 3 Faculty of Clinical Sciences, Department of Nursing Sciences, College of Health Sciences, University of Ilorin, Ilorin, Nigeria, 4 School of Nursing and Midwifery at Queens University in Belfast, Belfast, Northern Ireland, United Kingdom, 5 Society for Family Health, Abuja, Nigeria, 6 World Health Organization, Abuja, Nigeria, 7 Plateau State College of Nursing Sciences, Vom, Plateau State, Nigeria

* hwada@sfhnigeria.org

**Data Availability Statement:** All relevant data will be available with a substantial request to the data repository centre at the Society for Family Health

## Abstract

Healthcare workers continue to experience high levels of work-related stress which continue to negatively affect their psychological, physical, and emotional well-being. This is even more prevalent among healthcare workers who work in surgical specialities, with the surgical operation room becoming a known stressor at hospitals. This study aims to assess work-related stress among surgical team members at Ahmadu Bello University Teaching Hospital in Zaria between January 2021–2022. Data analysis involved descriptive and inferential statistical approaches using the Statistical Package for Social Science (SPSS) version 23.0. The study found an overall high prevalence of work-related stress, with 65% of participants reporting moderate levels of stress. The majority of the participants have a mean age of 39.4 ± 7.8 years, most of them being physicians (66.3%), being males (59.9%), and identified with a Hausa ethnic tribe. Notably, the multiple regression analysis found that tribe ($p = 0.008$), professional cadres ($p = 0.001$) and age/years of experience ($p = 0.0035$) emerged as significant predictors of work-related stress. Key determinants of work-related stress among surgical team members include workload, complexity of work, and conflicting cognitive job demands that continue to subject professionals to increasing workloads and constant decision-making about their job. Organizational factors, such as job policy and procedure, communication problems, and the nature of facilities, were identified as the highest contributors to work-related stress in organizational, interpersonal, and physical/environmental dimensions. The findings lead to the conclusion that a considerable proportion of surgical team members experience a relatively high level of work-related stress, primarily attributed to workload and cognitive demands. In light of these results, urgent efforts are recommended to improve the working conditions and environment for surgical team members. Furthermore, the integration of stress management measures into the educational

(SFH) at info@sfhnigeria.org or directly through the corresponding author at hwada@sfhnigeria.org or hasawa2011@gmail.com. This data contains potentially identifying or sensitive surgical workforce information and can be used against identified personnel. This have been protected and has been enshrined according to the Health Research Ethics Committee of ABUTH, Zaria (ABUTH 954524802) obtained before the start of the study.

**Funding:** The authors received no specific funding for this work.

**Competing interests:** The authors have declared that no competing interests exist.

programs for the surgical team is emphasized to effectively address and mitigate the impact of work-related stress.

## Introduction

The shortage of health professionals is getting more serious, and healthcare systems in several countries are struggling to attract a trained health workforce [1]. The situation is exacerbated in low-middle-income countries (LMICs) such as Nigeria, where a large number of health professionals migrate to high-income countries (HICs) in search of greener pastures [2]. The shortage is exacerbated further by the fluctuating economic conditions prevalent in the majority of LMICs, which make it difficult for employers to hire health professionals [3]. The few remaining health professionals are left to deal with an influx of people seeking medical attention. This can be detrimental to the emotional and physical health of health professionals [2]. Work-related stress is a key contributor to the decision of health professionals to migrate or end their careers prematurely [4]. According to the World Health Organization (2020), "work-related stress is the response people may have when presented with work demands and pressures that are not matched to their knowledge and abilities and which challenge their ability to cope" [5]. Stress at workplace continues to gain attention and have been recognized as a global disease due to its negative impact on the psychological, emotional, and physical well-being of people in different occupational groups [6,7]. Stress experienced by health professionals on the job continues to have a serious impact on general well-being including detrimental effects on their mental, emotional, physical, and social health, job satisfaction, and workplace safety, among other things [8]. They are confronted by an increasingly complex environment with heavy workloads, scarce resources, inadequate staffing, and expanding roles that continue to promote work-related stress [9]. Around 100 million workdays are lost each year as a result of stress, and stress is considered to be responsible for 50per cent to 75per cent of disease [10]. Employers are estimated to pay between $200 and $300 billion in expenditures associated with decreased performance, productivity, and quality, more accidents and injuries, increased healthcare costs, and increased absenteeism and turnover [11]. In Nigeria, evidence has shown that the country has a challenge of non-adequate surgical teams and an increasing challenge of surgical residency training [12,13]. The studies shows that issues relating to lack of funding of surgical trainees, inadequate trainees, low motivation workplace, bullying, and high clinical load continue to hamper surgical activity in Nigeria [12,14,15]. Work-related stress has been linked to cardiovascular disease, musculoskeletal discomfort, and depression [16]. Healthcare workers, particularly those in perioperative care, tend to be more vulnerable to stress [17]. However, studies have reported that the surgical workforce is the most susceptible to work-related stress, which might worsen outcomes for patients and providers [18,19]. Therefore, understanding the dynamics of the risk factors associated with the stress may provide a more specific and contextual strategy for preventing and treating them. Moreover, a study has also shown that approximately half of surgical team members experienced moderate to severe stress as a result of long hours spent in operating rooms [20]. Understanding the prevalence and associated factors related to these levels of stress is key for decision-making. It offers an opportunity to develop more tailored measures of workforce development and health promotion [18]. Stress is also increasingly recognized as a contributor to poor teamwork in the operating room in Nigeria, implying that high levels of stress are deleterious to team performance [21]. This may lead to professional consequences (poor patient satisfaction, low quality of care, or medical errors), all of which could result in malpractice suits costing caregivers and hospitals large sums of money [21]. Similarly, this may have severe personal repercussions such as

substance abuse, broken relationships, and even suicide [22]. Reducing exposure to significant stressors at work and improving working conditions therefore critical for minimizing work-related stress among surgical team members [23]. As a result, identifying the most significant stressors in the workplace is critical for designing effective preventative measures [24]. Although numerous studies on this subject have been undertaken in HICs, there is still a dearth of studies from LMICs including Nigeria [25,26]. Additionally, due to socioeconomic and cultural considerations, findings from HICs may not apply to LMICs. Therefore, this study determines the prevalence of perceived work-related stress among surgical workforce, identifies determinants of perceived work-related stress, and evaluates the association between perceived work-related stresses among the cohort.

## Methods

### Study setting and design

A descriptive cross-sectional study was conducted from January 2021 to January 2022 within the operating theatres of Ahmadu Bello University Teaching Hospital (ABUTH) in Zaria, Nigeria. The surgical specialties included General Surgery, Ophthalmic Surgery, Maxillofacial Unit, Urology, Plastic and Reconstruction, Cardio-Thoracic, Ear-Nose and Throat, Paediatric Surgery, Orthopaedic, Neurosurgery, and Obstetrics and Gynaecology. The participants comprised Perioperative Nurses, Nurse Anesthetists, Physician Anesthesiologists, Surgeons, and Registrars, all integral members of the surgical workforce team. The department receives an average of 1500 elective surgical admission (ELA) and over 700 emergency surgeries in a year.

ABUTH is a 500-bed tertiary care institution located in Zaria City, Kaduna State, Northwest Nigeria, and serves as a teaching hospital for Ahmadu Bello University [27]. The hospital has a surgery department that provides surgical support and services to different specialist subunits including ophthalmic, maxillofacial, urology, plastic, cardio-thoracic, paediatric, orthopedic, gastrointestinal neurology, obstetrics, and gynecology units.

### Inclusion and exclusion criteria

We included surgical team members who had at least six months of experience working in the operating rooms at ABUTH, Zaria (the six months of experience was to reduce social desirability bias which could result in overrepresentation of the responses of respondents who might likely not have had any experience of performing surgical procedures and a major requirement for civil service employment to be on probation). Excluded from the study were all locum theatre employees, students on clinical posting to the theatres, and those who were unwilling to participate. The reason for excluding tenure appointments and students on clinical posting to the theatre was to avoid confounding factors of the result, which stress-related might have been due to hangovers due to late-hour studies, travels relating to locum staff, academic-related questions being asked and prior stress that they might have had not related to the surgical operations being done. Further, locum staff and students are not recognized as staff of the institution and students were not allowed to perform any surgical procedures.

### Sample size and sampling technique

A minimum sample size of 189 surgical team members was established using Yamane's formula ($n = N/ 1+N (e)^2$) [28]. The surgical team was recruited using a stratified sample procedure from two strata: consultant surgeons, anaesthesiologists, and registrars in one stratum. Another stratum includes perioperative nurses and nurse anesthetists. The proportion of surgical team members to be selected per stratum was determined using the formula $n_1 = n/NXS$.

Where:

$n_1$ = number of respondents to be selected from each cadre

n = total number of surgical teams in each cadre (physicians, nurses).

N = total number of surgical team members in ABUTH

S = minimum sample size determined.

## Instrument

A structured, self-administered survey was employed for data collection, organized into four sections. The first section focused on socio-demographic information, encompassing gender, age, tribe, religion, occupation, years of experience in the operating room, marital status, education, monthly income, average working hours per week, and surgical team specialty.

The second section incorporated the Perceived Stress Scale-10 (PSS-10) [29], a widely used global measure of perceived stress known for its reliability in predicting health outcomes [30]. Utilized in various settings, including the United Kingdom [31] and India [32], the PSS-10 assessed perceived stress over the past 30 days on a 5-point scale (0 = Never, 1 = Almost never, 2 = Sometimes, 3 = Fairly Often, 4 = Very Often). The total scores, ranging from 0 to 40, demonstrated a high internal consistency reliability (α = .84).

The third section evaluated the determinants of perceived work-related stress, utilizing a questionnaire adapted from the National Institute of Occupational Safety and Health (NIOSH) Generic Job Stress Questionnaire [33]. All questions were equally weighted, exploring participants' agreement or disagreement with factors categorized as job, organizational, interpersonal, and working environment related. Responses were measured on a five-point Likert scale (4 = Strongly Agree to 0 = Strongly Disagree), with participants who scored strongly agree and agree categorized as agreeing, and those who scored neutral, disagree, and strongly disagree categorized as disagreeing.

The final section utilized a researcher-designed questionnaire with 13 questions to measure the outcome variables of work-related stress among surgical team members. These variables were continuous and reported as highly agree, agree, neutral, disagree, and strongly disagree. Participants who scored neutral, disagree, and strongly disagree were categorized as disagreeing, while those who rated strongly agree and agreed were categorized as agreeing.

**Validity and internal consistency.** The survey underwent a pilot test–in which both content and face validity, and reliability were done using Cronbach's alpha analysis. A pre-test was performed to assess the face and content validity on 10% of the sample, drawn from a similar group in a different institution (Barau Dikko Specialist Hospital, Kaduna) which contains both surgical consultants, registrars on consultancy/specialist training, surgical nurses, and medical officers who practice in the surgical theatre to assess the face validity. They were requested to comment on the difficulty level and assess whether each item was clear (understandable), relevant (important), and essential (necessary). Since the majority of the survey instrument was adapted, we used the preliminary findings of the validity to assess the relevancy and the flow of the questions. Based on the same preliminary findings, minor adjustments were made to enhance the survey. Subsequently, to establish the quantitative validity and reliability of the instrument, Cronbach's alpha was employed. The Cronbach's alpha for the Perceived Stress Scale-10 (PSS-10) ranged from 0.76 to 0.88, indicating satisfactory internal consistency. For the determinant of perceived work-related stress, the alpha ranged from 0.62 to 0.86, and for the outcome variable of work-related stress, it ranged from 0.67 to 0.81, further affirming the reliability of the instrument. The final research questionnaire used is attached as a supporting document [S1 File].

**Data collection.**   Four research assistants, all of whom had received training before data collection, were responsible for data collection and they collected the data using a paper-based printed questionnaire. This training took place over three consecutive days at the ABUTH Modular Theatre Lounge. The pilot tests which were discussed in the validity section were done by the principal investigator (PI) and were excluded in any analysis reported in this research. The questionnaire administration was done based on workforce allocation [Table 1] and done after surgical procedures to ascertain the true level of stress they had. Participants on the surgical teams were approached from each operating room (chosen purposively) and given a survey if they agreed to take part in the study. However, whenever a questionnaire was not filled (especially due to stress or had patient reviews to undergo), we made follow-up visits to ensure that completion of all sections was made after work time.

**Data analysis.**   The collected data underwent analysis using the Statistical Package for the Social Sciences (SPSS), version 23.0. Univariate analysis was employed for descriptive statistics, including frequencies, percentages, mean, and standard deviation. The relationship between dependent and independent variables was explored through bivariate analysis, utilizing tests such as the Chi-square test, student t-test, one-way ANOVA, and Mann-Whitney U test. Predictor variables displaying a significant association with the outcome variables underwent further scrutiny through multivariate analysis, specifically multiple regression. This analysis aimed to identify factors that predispose participants to work-related stress. Odds ratio estimates greater than 1.0 were used to indicate that the odds of the event happening were higher than that for the reference category, and vice versa. For all tests, a statistical significance level of $P \leq 0.05$ was applied, and the tests were two-tailed.

**Ethical consideration.**   Ethical clearance was obtained from the Health Research Ethics Committee of ABUTH, Zaria before the commencement of the study (ABUTH 954524802). Informed Consent for Participation: Written consent was obtained from the participants and all information from the participants was kept strictly confidential. No part of this study provided any potentially identifying information about participants. The ethical clearance and other documentations are attached as a supporting document.

**Inclusivity in global research.**   Information regarding the ethical, cultural, and scientific considerations specific to inclusivity in global research is included in the [S2 File].

## Results

The result following the data collection shows that out of 189 surgical team members who were recruited for this study, only 177 personnel fully completed the survey process (93.7% completion rate).

### Socio-demographic characteristics

As presented in Table 2, the participants in this study had a mean age of 39.4 ± 7.8 years, with ages ranging from 21 to 51+ years. The majority of participants were physicians (n = 116; 65.5%), and males constituted (n = 106; 59.9%) of the sample. More than half of the participants, accounting (n = 95; 53.7%), identified as Muslims. The ethnic distribution reflected

**Table 1. Proportion of surgical team members to be selected per strata.**

| Cadre | Total number of each cadre in hospital | Total number selected from each cadre |
| --- | --- | --- |
| Doctors | 187 | 142 |
| Nurses | 61 | 47 |
| **Total** | **246** | **189** |

**Table 2. Socio-demographic characteristics of the respondents (n = 177).**

| Variables | Frequency | Per cent |
|---|---|---|
| Gender | | |
| Male | 106 | 59.9 |
| Female | 71 | 40.1 |
| | | |
| Age group (years) | | |
| 21–30 | 32 | 18.1 |
| 31–40 | 73 | 41.2 |
| 41–50 | 63 | 35.6 |
| 51 and above | 9 | 5.1 |
| | | |
| Tribe | | |
| Hausa/Fulani | 48 | 27.1 |
| Yoruba | 25 | 14.1 |
| Igbo | 20 | 11.3 |
| Others | 84 | 47.5 |
| | | |
| Religion | | |
| Islam | 95 | 53.7 |
| Christianity | 82 | 46.3 |
| | | |
| Years of work experience in the operating theatre | | |
| 0–4 years | 75 | 42.4 |
| 5–9 years | 64 | 36.2 |
| 10–14 years | 26 | 14.6 |
| 15–19 years | 8 | 4.5 |
| 20–24 years | 1 | 0.6 |
| 25–29 years | 0.0 | 0.0 |
| 30–34 years | 3 | 1.7 |
| Mean year of work experience | 6.0±5.4 | |
| | | |
| Cadre | | |
| Doctors | 116 | 65.5 |
| Nurses | 61 | 34.5 |
| Highest level of education | | |
| PhD | 1 | 0.6 |
| Fellowship | 39 | 22.0 |
| MSc/Masters | 22 | 12.4 |
| PGD | 15 | 8.5 |
| BSc. | 44 | 24.9 |
| HND/Equivalent | 5 | 2.7 |
| Diploma | 7 | 4.0 |
| Others | 38 | 21.5 |
| No response | 6 | 3.4 |
| | | |
| Marital status | | |
| Single | 60 | 33.9 |
| Married | 110 | 62.1 |
| Separated | 1 | 0.6 |
| Divorcee | 5 | 2.8 |
| Widow | 1 | 0.6 |

**Table 3. Perceived prevalence of level of work-related stress among respondents (n = 177).**

| Level of stress | | Frequency | Per cent |
|---|---|---|---|
| | Lack of stress | 5 | 2.8 |
| | Low stress | 21 | 11.9 |
| | Moderate stress | 115 | 65.0 |
| | Severe stress | 36 | 20.3 |

Hausa (n = 48.2; 27.1%), Yoruba (n = 25; 14.1%), and Igbo (m = 20; 11.3%) as the predominant groups. Notably, Table 2 highlighted that 42.4% of the participants had between 0 and 4 years of experience working in the operating room.

**Prevalence of level of work-related stress.** The majority of participants (n = 115;65%) reported moderate levels of work-related stress, whereas (n-36; 20.3%) reported severe stress, and (n = 21; 11.9%) reported low levels. This indicates that most surgical team members regard their work as stressful (see Table 3).

**Perceived determinants of work-related stress.** Workload and task complexity (mean 4.060.77 and 3.970.69, respectively) were the most influential determinants of perceived job-related stress, while role ambiguity was the least determinant (mean 3.860.8SD) (see Table 4). As a result, the most common work-related stressors in the operating room are workload, task complexity, and conflicting job demands (cognitive).

**Relationship between perceived work-related stresses and cadre of team members.** The study findings indicated that both physicians and nurses perceived work-related stress significantly, with physicians recording a higher mean value of work-related stress than nurses (refer to Table 5). Importantly, there was a notable correlation (p = 0.000) between the prevalence of perceived work-related stress, exposure to categories of work-related stress, and the professional cadre of the participants. Both professional groups displayed similar mean values for work-related stress in the operating theatre, organizational-related stress, and interpersonal-related stress, highlighting significant differences between physicians and nurses.

In Table 6, the multiple regression analysis aimed at predicting the seven factors of work-related stress revealed that the coefficient of the multiple regressions (R) was 39.0%, indicating a good level of prediction. The R-squared value (0.157) represented the proportion of variation

**Table 4. Perceived determinants of work-related stress (n = 177).**

| Causes | SA* | A* | N* | D* | SD* | Mean ± SD | Rank |
|---|---|---|---|---|---|---|---|
| Lack of support from seniors | 28 | 119 | 15 | 11 | 4 | 3.88±0.83 | 7 |
| Negative experience with patients/ relatives | 24 | 128 | 15 | 7 | 3 | 3.92±0.73 | 4 |
| Complexity of work | 27 | 127 | 15 | 6 | 2 | 3.97±0.69 | 2 |
| Conflicting job demands (cognitive) | 38 | 107 | 20 | 11 | 1 | 3.96±0.79 | 3 |
| Workload | 51 | 91 | 29 | 6 | 0 | 4.06±0.77 | 1 |
| Role conflict | 48 | 81 | 32 | 14 | 2 | 3.89±0.93 | 6 |
| Lack of balance between personal and work life | 44 | 94 | 25 | 9 | 5 | 3.92±0.92 | 4 |
| Role ambiguity | 41 | 86 | 35 | 14 | 1 | 3.86±0.88 | 8 |
| Dealing with death and dying | 50 | 80 | 29 | 18 | 0 | 3.91±0.92 | 5 |
| Inadequate preparations of patients for surgery | 48 | 88 | 23 | 14 | 4 | 3.92±0.96 | 4 |
| **Cumulative mean** | | | | | | **3.929** | |

Standard/decision mean = 3.00.

*SA = Strongly agreed; *A = Agreed; *N = Neutral; *D = Disagreed; *SD = Strongly Disagreed.

**Table 5. Correlation between perceived work-related stresses with cadre.**

| Variables | Cadre | Mean | SD | Sig. | Remarks |
|---|---|---|---|---|---|
| Prevalence of perceived work-related stress | Doctors | 34.3707 | 4.42037 | 0.000* | P<0.05 |
| | Nurses | 31.2881 | 4.80320 | | |
| Exposure to categories of work-related stress | Doctors | 17.4397 | 2.55815 | 0.000* | P<0.05 |
| | Nurses | 15.7119 | 2.98300 | | |
| Factors contributing to work related stress | Doctors | 39.7826 | 5.24117 | 0.114 | p>0.05 |
| | Nurses | 38.3448 | 6.29551 | | |
| Organizational related stress | Doctors | 12.0431 | 1.42275 | 0.399 | p>0.05 |
| | Nurses | 12.2373 | 1.46616 | | |
| Interpersonal related stress | Doctors | 16.0517 | 2.62047 | 0.608 | p>0.05 |
| | Nurses | 15.2542 | 2.89228 | | |

in the dependent variables in the regression model, suggesting a strong predictor of work-related stress among surgical team members. Furthermore, the results demonstrated that the overall calculated p-value of 0.000 was lower than the 0.05 level of significance, and the computed F value of 4.509 exceeded the 3.000 F critical value. Consequently, tribe, professional cadre, and age collectively predicted work-related stress among the surgical team members.

## Discussion

The study findings reveal that a little over two-thirds (65%) of the respondents experienced a moderate level of work-related stress, indicating a notable prevalence of such stress in the operating theatre setting. Previous research has reported a slightly lower prevalence (51%), suggesting that work stress levels may not be decreasing in the country despite similar settings and work environments [34]. Interestingly, our study might have faced an over-or-underestimation of work-related stress based on our survey, based on the reality that their present subjection to the work-related stress they had at the time and may feel empowered to fill the survey, whereas those who are not subject to work-related stress may not feel the need to fill in the survey. Conversely, many of the surgical workforce might have not been comfortable filling in the survey due to fear of repression by the hospital. However, we were able to reduce this bias by removing any information that would identify any professional including disaggregating data by department of specialty. As such, we identified a sample based on the willingness

**Table 6. Multiple regression analysis between independent variables and prevalence of work-related stress among surgical team members.**

| Variables | Coefficient | Beta | t-cal | Sig. |
|---|---|---|---|---|
| Tribe | -.742 | .202 | 17.808 | 0.008* |
| Cadre | -3.393 | -.352 | -2.664 | 0.001* |
| Educational Qualification | .060 | .029 | -3.460 | 0.707 |
| Marital status | .849 | .119 | .376 | 0.122 |
| Specialties | .325 | .109 | 1.554 | 0.311 |
| Monthly income | .211 | .060 | 1.017 | 0.425 |
| Age | 1.015 | .176 | .800 | 0.035* |
| **Constant** | **34.505** | | **17.808** | **0.00** |
| **R = 0.390[a]** | | | | |
| **R2 = 0.157** | | | | |
| **R2 (adj) = 0.122** | | | | |
| **F- ratio = 4.509** | | | | |

to participate and the diversity of the professionals which provides a baseline idea and information regarding perceived stress in the surgical workforce [35]. The stressful nature of the environment in this surgical workforce exposes them to higher risks, as found in this study and others, including long working hours, musculoskeletal pain, and fatigue. The study identifies organizational/institutional and physical factors as the most frequently perceived work-related stressors among operating team members. Consistent with previous reports, physical, organizational, and other related hazards in operating theatres are highlighted [36,37]. Perceived causes of work-related stress in this study include workload and complexity of work, conflicting cognitive job demands (ever-changing working conditions that subject workers to increasing deadlines and workloads, constant planning and decision-making about one's career and job), lack of balance between personal and work life, and dealing with death and dying, all ranking high. This finding aligns with reported studies that show associated long working hours, excessive workload, and dealing with death and dying with work-related stress [38,39]. Similarly, studies have suggested that time/work pressure and increased workload are significant factors causing work-related stress in the operating room [40–43]. The implication is that pressure from work and the intricacy of tasks are major predictors of work-related stress in the operating theatre.

A study has reported that these negative pressures have continued to lead nurses to low psychological and physical well-being leading many towards leaving the profession or moving to other countries with less pressure [44]. Our multiple regression analysis identified professional cadre, age, and tribe as significant predictors of perceived work-related stress. These findings are compatible with similar studies assessing the surgical workforce. For instance, a study in Malawi identified a relationship between sources of stress and demographical variables in nurses [45]. They also reported a significant relationship between age, organizational-related stressors, marital status, and educational qualification with work-related stress. On the other hand, evidence has shown a significant association between factors like income status and stress scores. At the same time, another study from Iran found no significant difference in the prevalence of work-related stress among surgical teams among different variables such as years of experience, number of dependents, and average working hours [46]. This differs from the findings of on real-time observations of stressful events in the operating room, which reported no significant association between stress levels and age, gender, working shift, and the number of dependents in Saudi Arabia [47]. However, they found a significant inverse association between stress levels and years of work experience. In this study, professional cadre and age were factors for work-related stress. Accordingly, this is in line with studies done in Ethiopia [48–50], and Botswana [51] which reported that younger nurses and were at a lower professional cadre were less stressed than those who were aged and experienced. One possible explanation for this might be due to physiological factors and the higher roles of aged and experienced professionals in the hospital [49]. However, this trend was contradictory in a study done in Singapore, where surgical nurses who were aged and experienced were more stressed [52]. This is best explained due to sociodemographic differences and the study being conducted in a private hospital, whereas our own was done in a public teaching hospital where higher cadre professionals provide teaching guidance to different sets of students and trainees.

## Study recommendation

This study has several public health implications including providing evidence that will optimize adaptation and mitigation strategies of human resources for health amid inadequate resources. The majority of the stressors are institutional or organizational based which is consistent with a similar study in Ghana, where there are challenges relating to inadequate

resources, absence of work balance, workload, and opportunity to grow [53]. This possibly suggests that there are inadequate human resources or an absence of job duties to carry out their respective roles and responsibilities, in such a way as to reduce stress. A vital recommendation for policymakers is ensuring is the need for increased number human resources for health in surgical work reduced workload and having a defined time role, particularly for residents who are trainees. This will require more quotas for residents in tertiary hospitals and to provide adequate renumeration to the overstretched workforce in delivering better quality care and reducing stress. Secondly, as Nigeria defines its human resource for health policy framework in the Nigeria Health Sectoral Renewal Investment Initiative–aimed at delivering improved quality health by retraining human resources for health and improving the number of healthcare workers in healthcare centers, the Nigerian government should look at innovative ways to incentivize overtime and long shift for younger doctors. In a recent study in Nigeria, the mean working hours per week was 71.79 hours per week among early-career surgeons [54] and 122.7 hours for resident doctors [55]. This is above the 48 hours per week as recommended by the European Union (EU) and the maximum stipulated 80 working hours per week by the ACGME and Institute of Medicine (IOM) [56,57]. However, there are no penalties or incentives for long working hours in complying with the prescribed hours. Thirdly, our study found that age group, professional cadre, and tribe are associated stressors, which can negatively impact the quality of caregiving. This is consistent with findings from a study in a Dutch hospital, which also highlighted these factors as significant stressors [58]. In Nigeria, professional cadre tends to have major implications on associated stress, whereby consultants tend to expect resident or junior surgeons to be proficient in surgical skills and surgical experiences, leaving them to attend to the majority of cases with major burnouts and stress [14]. This is seen as a critical gap that exists in the country's surgical human resources with a reported 1.8 per 100,000 population vs surgical specialist density in Nigeria in comparison with the Lancet Commission's recommendation of 20 per 100,000 [59]. Similar to this, studies have reported consistent findings with surgical nurses, in which junior nurses in the surgical specialty tend to have high burnout due to high workload, low control over their job, and low rewards [60,61]. In a similar vein, younger professionals including the surgical workforce tend to confide with senior colleagues who are of the same tribe to provide them with advice and other measures to counteract those feelings, create more balance, and alleviate stress [58]. We equally hope that the government and relevant regulatory professions will look at instituting stress inoculation and management programmes for doctors and other health professionals to enable to seek professional advice and how to mitigate the negative consequences associated with distress and burnout.

## Limitation

The study despite its strength has some limitations, particularly about its cross-sectional nature which could not provide a temporal relationship between the independent and outcome variables. Secondly, the non-proportional distribution and the sapling technique between the study participants (doctors vs nurses) might mask the opinion of the nurses. This this a single-center study, to which the study findings have limited generalizability. However, it highlights a significant and pressing issue that requires attention and can inform policymakers in enacting change. Further, our study was carried out at the end of the COVID-19 pandemic with ongoing consequences of backlog generated during the pandemic. This can be a confounding factor on the stress variable, of which the effect was not investigated. We recommend that studies in various hospitals across Nigeria should be carried out to gain a comprehensive understanding of the national scenario. Our study might have a difference of socio-cultural and health

systems settings between studies within HIC, which might be attributed high prevalence of stressors with the same measurement tool. Despite this, the study provides a baseline assessment and evidence for policymakers, governments, institutions, and medical regulators, that can be used for policy intervention and workplace balance.

## Conclusion

The study findings reveal that a little over two-thirds (65%) of the respondents experienced a moderate level of work-related stress, indicating a notable prevalence of such stress in the operating theatre setting. Our multiple regression analysis identified professional cadre, age, and tribe as significant predictors of perceived work-related stress among our study cohorts. The findings lead to the conclusion that a considerable proportion of surgical team members experience a relatively high level of work-related stress, primarily attributed to workload and cognitive demands. In light of these results, urgent efforts are recommended to improve the working conditions and environment for surgical team members including targeted interventions such as social training, capacity building, workflow improvement, supportive team-based approaches, regular workplace social hours, flexible working hours, interpersonal communication chains and anonymous talk-it out mechanisms. Furthermore, the integration of stress management measures into the educational programs for the surgical team is emphasized to effectively address and mitigate the impact of work-related stress.

## Supporting information

**S1 File. Questionnaire.**
(PDF)

**S2 File. Inclusivity in global research.**
(PDF)

## Author Contributions

**Conceptualization:** Jibril M. Bashar, Danjuma Aliyu, Emmanuel E. Anyebe, Israel Gabriel, Yusuf H. Wada.

**Data curation:** Jibril M. Bashar, Yusuf H. Wada.

**Formal analysis:** Jibril M. Bashar, Israel Gabriel, Abdulrrahaman S. Mangari, Yusuf H. Wada.

**Investigation:** Emmanuel E. Anyebe, Israel Gabriel, Amanullahi Nasir.

**Methodology:** Danjuma Aliyu, Emmanuel E. Anyebe, Yusuf H. Wada.

**Project administration:** Jibril M. Bashar, Israel Gabriel, Amanullahi Nasir, Abdulrrahaman S. Mangari.

**Resources:** Danjuma Aliyu, Faizah S. Abubakar.

**Software:** Israel Gabriel, Faizah S. Abubakar.

**Supervision:** Jibril M. Bashar, Emmanuel E. Anyebe, Israel Gabriel.

**Validation:** Danjuma Aliyu, Emmanuel E. Anyebe, Faizah S. Abubakar.

**Visualization:** Amanullahi Nasir, Yusuf H. Wada.

**Writing – original draft:** Jibril M. Bashar, Emmanuel E. Anyebe, Israel Gabriel, Yusuf H. Wada.

**Writing – review & editing:** Jibril M. Bashar, Danjuma Aliyu, Israel Gabriel, Amanullahi Nasir, Abdulrrahaman S. Mangari, Faizah S. Abubakar, Yusuf H. Wada.

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
