## [Decision Letter · Decision Letter 0]

15 Jul 2024

PGPH-D-24-01269

Work-related stress and associated factors among surgical team members in Nigeria’s operating room: A cross-sectional study

Dear Dr. Hassan Wada,

Thank you for submitting your manuscript to PLOS Global Public Health. After careful consideration, we feel that it has merit but does not fully meet PLOS Global Public Health’s publication criteria as it currently stands. Therefore, we invite you to submit a revised version of the manuscript that addresses the points raised during the review process.

We look forward to receiving your revised manuscript.

Kind regards,

Isabella Faria, MD

Academic Editor

Journal Requirements:

2. In this instance it seems there may be acceptable restrictions in place that prevent the public sharing of your minimal data. However, in line with our goal of ensuring long-term data availability to all interested researchers, PLOS’ Data Policy states that authors cannot be the sole named individuals responsible for ensuring data access (http://journals.plos.org/plosone/s/data-availability#loc-acceptable-data-sharing-methods).

Additional Editor Comments (if provided):

The subject of research is important and understudied and the methodology is sound. The paper can be greatly improved with the feedback provided by our reviewers. I agree with both reviewers that the manuscript should be reviewed by either a native English speaker before sending the revised version to improve clarity and information flow. I commend the authors for the important work and look forward to reading the revised version.

Reviewers' comments:

Reviewer's Responses to Questions

**Comments to the Author**

Reviewer #1: Thank you for allowing me to review this work. I commend the authors on addressing such an important topic. As highlighted in this manuscript, it is crucial to protect the mental health of the health workforce, with work-related stress being an important factor. This issue is particularly pressing in LMICs, where it is still underreported due to stigma and other barriers.

Overall, the findings are compelling; however, I recommend substantial revisions, with particular attention to significant rewriting.

Title:

1. Several statements, including the title, imply causation and national impact. “Work-related stress and associated factors among surgical team members in Nigeria’s operating room: A cross-sectional study”

- I suggest adding words such as "Perceived" to mitigate the implication of causation.

- Instead of "surgical team members," I suggest using "Surgical Workforce" and maintaining this terminology consistently throughout the text.

- Be cautious with the term "in Nigeria’s operating room" Since the study is based on a single center with a limited cohort, it cannot be generalized to other facilities in Brazil. Consider rephrasing to reflect this limitation.

Abstract:

1. I would suggest the authors add numerical results in the abstract for clarity.

2. Include the time period during which the study was conducted to provide context.

3. Suggested sentence: "The study found an overall prevalence of work-related stress, with 65% of participants reporting moderate levels of stress."

4. Clarify the sample size (189 or 177?). Ensure consistency in the reported number of participants.

Introduction:

1. While the introduction appears to be sound, the writing is unclear, making it difficult to follow. I advise the authors to work with a writing coach or copyeditor to improve the flow and readability of the text. This also applies for the discussion section.

2. There is no need to define surgical workforce in the introduction as it is now: “The perioperative teams consist of several kinds of professionals, including surgeons, anaesthesiologists, nurses, and surgical technicians [18]. Nurses working in the operating room include scrub and circulating nurses, as well as nurse anaesthetists [19].” In any case this should be moved to the methodology section to define what was considered by the authors as an inclusion criteria for this study.

Methods:

I commend taking into consideration different specialties and different workforce, as well as using standardized text and rigorous methodology. However, there are some points to be clarified:

1. Please cite the “National Institute of Occupational Safety and Health (NIOSH) Generic Job Stress Questionnaire.”

2. Is the survey available? This is important for reproducibility and to further understand how the questionnaire was developed and the questions asked.

3. It would be helpful to provide a brief background on the country’s surgical training, surgical work process, and workload, either in the “Study Setting and Design” section or in the discussion. Expand on the explanation of this hospital’s structure/types and compare it to other hospitals in Nigeria. Does it have a higher volume? Higher work hours? Less staff? What are the characteristics of this hospital and setting? This would help in understanding the results and their association with work-related stress.

4. Can the authors elaborate on the selection bias of their study sample? This is briefly acknowledged but not discussed in detail in the limitations. For example, how might the survey length (many questions), the topic (mental health, work-related stress), and other commitments (as work tasks and work volume are important issues) influence the results?

5. Could the authors expand on how this test was performed and how the questions were modified? Based on what? Was it through cognitive tests, pilot tests, or focus groups? Did respondents share what they understood or not? How was the process documented and what needed to change? “The survey underwent a pre-test on 10% of the sample, drawn from a similar group in a different institution (Barau Dikko Specialist Hospital, Kaduna). Based on the preliminary findings, minor adjustments were made to enhance the survey.”

6. I would suggest the authors expand on the process “ A descriptive cross-sectional study was conducted from January 2021 to January 2022 within the operating theatres of Ahmadu Bello University Teaching Hospital (ABUTH) in Zaria, Nigeria.” Who delivered the surveys? Were they digital or paper-based? How frequently were they administered? How much time did it take to complete each survey? Was it before or after the surgical procedures? Before or after the staff shift? It would be important to understand more.

Results:

1. The results in the table do not sum up correctly:

177 for gender, 178 for age, and 175 for cadre.

2. Please standardize how numbers are reported.

For instance: “ The majority of participants (n=115, or 65per cent) reported moderate levels of work-related 214 stress, whereas 20.3 per cent reported severe and 11.9 per cent reported low levels, 215 respectively” then “The majority of participants were physicians (n=116; 66.3%), and males constituted 59.9% (n=106) of the sample. More than half of the participants, accounting for 53.7% (n=95), identified as Muslims. The ethnic distribution reflected Hausa (48.2%), Yoruba (14.1%), and Igbo (11.3%)

3. As different specialties were collected, it would be interesting to have the distribution reported. Have the authors assessed the difference in stress burden among these specialties?

4. What was the completion rate of the survey? How were incomplete responses handled?

5. The terms "conflicting job demands (cognitive)" and "Organizational factors, such as job policy and procedure" are unclear, please clarify what they mean. Overall the categories could benefit from more detail on what they encompass and how they were explained to respondents.

Discussion/Limitations:

The discussion brings many important insights, reflecting on the manuscript's findings and comparing them with existing literature. However, I would like to bring up some points that can improve this section:

1. Can the authors expand on what could be the reasons behind their findings? For instance, on the predictors of perceived work-related stress among surgical team members.

2. Enhance the clarity of your sentences and paragraphs by breaking them down into simpler, more concise statements.

3. The sentence “A vital recommendation for policy makers is ensuring that there is an increased training for human resources for health in surgical work, reduced workload and having a define time role particularly for residents as Nigeria define its human resource for health policy framework in the Nigeria Health Sectoral Renewal Investment Initiative.” is a clear example of how it would benefit on breaking it down. Additionally, Please expand on this Policy. How many hours are recommended? By how much does Nigeria usually comply? What are the penalties of not complying to this law?

4. On this sentence “This contrasts with the findings that showed overall low levels of perceived work-related stress, suggesting a potential steady increase in stress levels in the operating theatre” Authors must be cautious, as it is not appropriate to infer an increase in work-related stress across the country based on a single report from another hospital in a different region and time. I suggest removing this statement from the manuscript or revising it to indicate that it would be valuable to assess the situation over time at this center and other hospitals in Nigeria to detect an increase in work-related stress nationwide. Additionally, an important aspect to consider is that this study was conducted in 2021, during which the COVID-19 pandemic significantly impacted the surgical workforce. The ongoing consequences of the backlog generated during the pandemic, which were not mentioned in the discussion of this manuscript, should also be addressed.

5. Please be clear when describing findings from your study and other studies. When referencing other studies, provide specific information about their origin. For example, instead of writing: “Multiple regression analysis is consistent with a study that identifies professional cadre, age, and tribe as significant predictors of perceived work-related stress among surgical team members [45]. They also reported a significant relationship between age, organizational-related stressors, marital status, and educational qualification with work-related stress [45].”

Consider writing: “Our multiple regression analysis identified professional cadre, age, and tribe as significant predictors of perceived work-related stress. These findings are compatible with similar studies assessing the surgical workforce. For instance, a study in Malawi identified a relationship between sources of stress and demographic variables in nurses. Another study from [location] found [specific findings]. Interestingly, a study from [location] found [other specific findings].”

Ensure similar clarity throughout the text. For example: “The associated stressors were found to be age group, professional cadre, and tribe, which can have negative consequences on the quality of caregiving [49].”

Should be clarified to: “Our study found that age group, professional cadre, and tribe are associated stressors, which can negatively impact the quality of caregiving. This is consistent with findings from a study in [location], which also highlighted these factors as significant stressors [49].”

6. Can the authors comment on whether there may be an over- or under-estimate of work-related stress based on their survey? In reality, there may be bias due to self-selection of individuals filling in the survey because they are subject to work-related stress and may feel empowered to fill in the survey, whereas those who are not subject to work-related stress may not feel the need to fill in the survey. Conversely, professionals subject to work-related stress may be uncomfortable filling in the survey, for example, due to fear of repression by the hospital.

7. It should be acknowledged in the limitations section that this is a single-center study. While it highlights a significant and pressing issue that requires attention and can inform policymakers in enacting change, the findings may not be generalizable to other settings. Therefore, further studies in various hospitals across Nigeria are necessary to gain a comprehensive understanding of the national scenario.

Conclusion:

1. Where is the conclusion of this article? The manuscript should include a clear and concise conclusion that summarizes the main findings, their implications, and potential future research directions.

Reviewer #2:

Dear Authors, Thank you for sharing your work titled “Work-related stress and associated factors among surgical team members in Nigeria’s operating room: A cross-sectional study.” The article focuses on a pertinent issue faced by health workers. The study methods are sound and align well with the conclusions drawn from the findings. Here are some recommendations for improving your manuscript:

 Introduction

The introduction has redundancies in the description of the problem. Streamlining this section will enhance readability and focus.

Further describe the rationale for selecting only surgical specialties. This will help readers understand the specific relevance and implications of the study.

Discuss any potential biases in your sampling technique to provide a balanced view of the study’s strengths and limitations.

Add a description of the patient volume in the surgical department. This information will better capture the current workload of the study participants and contextualize the stress levels reported.

Methods

The inclusion and exclusion criteria should be justified. Explain why a minimum of six months of experience is necessary and why locum theatre employees and students are excluded.

Detail the validation process of the survey instrument, including any pilot testing conducted to ensure reliability and validity.

Elaborate on the statistical tests used and their appropriateness for the data. Provide more context on the use of multiple regression analysis and how it enhances the robustness of your findings.

Data Stratification

Further stratify the data to report specific stressors associated with variables such as gender. This will help shape recommendations tailored to the target population and make the most of the study population.

Recommendations and Conclusions

Add a dedicated section for study recommendations and conclusions to summarize your findings. This will provide clear, actionable insights for readers.

Identify specific examples, if available, of successful efforts to mitigate health workers' stressors. This will guide readers on potential solutions and practical applications of your research.

Language and Grammar

The manuscript should be reviewed by a native English speaker to address minor grammatical and punctuation errors. This will improve the overall readability and professionalism of your paper.

---

## [Decision Letter · Decision Letter 1]

14 Oct 2024

PGPH-D-24-01269R1

Perceived work-related stress and associated factors among the surgical workforce in a Nigerian tertiary health facility: A cross-sectional study

Dear Dr. Hassan Wada,

Thank you for submitting your manuscript to PLOS Global Public Health. After careful consideration, we feel that it has merit but does not fully meet PLOS Global Public Health’s publication criteria as it currently stands. Therefore, we invite you to submit a revised version of the manuscript that addresses the points raised during the review process.

We look forward to receiving your revised manuscript.

Kind regards,

Isabella Faria, MD

Academic Editor

Journal Requirements:

Additional Editor Comments (if provided):

Thank you for the changes made to the manuscript. There a few small changes that we would like to recommend to strengthen the article. Thank you for your patience during the reviewing process

Reviewers' comments:

Reviewer's Responses to Questions

Reviewer #1: I would like to acknowledge and congratulate the efforts that the authors went to, in addressing all of the Reviewer's comments, mine included. The intent and impact of the paper has been significantly improved.

I believe this manuscript is important and I think other people would benefit from reading this paper in a publication.

Reviewer #2: Dear Authors,

Thank you for sharing a revised version of your work titled “Perceived work-related stress and associated factors among the surgical workforce in a Nigerian tertiary health facility: A cross-sectional study.” I thoroughly enjoyed reading it. Please find some comments for the text shared herewith:

Introduction

Line 75 page “I” needs to be in small case

Line 78 add comma between “low motivation workplace bullying”

Results

Please describe why tribe was selected as a variable by the authors, and how it relates in its significance as a stressor in the context of this study population.

Discussion

Line298 to 299, I believe the sentence is supposed to say “Conversely, many of the surgical workforce might have not been comfortable filling in the survey due to fear of repression by the hospital.”

In the discussion section, add numerical data from the studies cited to show the comparison between the present study and previous evidence.

Line 335, rephrase from “that nurses who had less age” to younger nurses

Recommendations

The recommendations section has a lot of overflow of points from the discussion. Addtionally, the recommendations are very generalized and need to be further tailored to the study populations.

A suggestion for the authors is to identify perhaps 3 main recommendations and share brief details on their execution. Include specific, actionable interventions that would help enhance the applicability of the recommendation to the study setting.

Conclusion

The authors are recommended to look further into smart experiments in how to reduce workload stress. It is worthwhile to discuss that normative measures of reducing health workforce stressors (more training, workplace social hours) go so far if there are not targeted interventions on improving workflow, work hours, interpersonal communication chains etc.

---

## [Editor Report · Decision Letter 2]

30 Oct 2024

Perceived work-related stress and associated factors among the surgical workforce in a Nigerian tertiary health facility: A cross-sectional study

PGPH-D-24-01269R2

Dear Mr Hassan Wada,

We are pleased to inform you that your manuscript 'Perceived work-related stress and associated factors among the surgical workforce in a Nigerian tertiary health facility: A cross-sectional study' has been provisionally accepted for publication in PLOS Global Public Health.

Best regards,

Isabella Faria, MD

Academic Editor

Thank you for the submission of this important work and for working on the suggestions. The work was significantly improved. I would like to congratulate the authors on such an important work.